# Recurrence Resonance and 1/*f* Noise in Neurons Under Quantum Conditions and Their Manifestations in Proteinoid Microspheres

**DOI:** 10.3390/e27020145

**Published:** 2025-02-01

**Authors:** Yu Huang, Panagiotis Mougkogiannis, Andrew Adamatzky, Yukio Pegio Gunji

**Affiliations:** 1Department of Intermediate Art and Science, School of Fundamental Science and Engineering, Waseda University, Tokyo 169-8555, Japan; kouiku921@gmail.com; 2The Unconventional Computing Laboratory, University of the West England, Bristol BS16 1QY, UK; panagiotis.mougkogiannis@uwe.ac.uk (P.M.); andrew.adamatzky@uwe.ac.uk (A.A.)

**Keywords:** recurrence resonance, part and whole, self-organized criticality, mutual information, quantum logic, proteinoid microshere

## Abstract

Recurrence resonance (RR), in which external noise is utilized to enhance the behaviour of hidden attractors in a system, is a phenomenon often observed in biological systems and is expected to adjust between chaos and order to increase computational power. It is known that connections of neurons that are relatively dense make it possible to achieve RR and can be measured by global mutual information. Here, we used a Boltzmann machine to investigate how the manifestation of RR changes when the connection pattern between neurons is changed. When the connection strength pattern between neurons forms a partially sparse cluster structure revealing Boolean algebra or Quantum logic, an increase in mutual information and the formation of a maximum value are observed not only in the entire network but also in the subsystems of the network, making recurrence resonance detectable. It is also found that in a clustered connection distribution, the state time series of a single neuron shows 1/f noise. In proteinoid microspheres, clusters of amino acid compounds, the time series of localized potential changes emit pulses like neurons and transmit and receive information. Indeed, it is found that these also exhibit 1/f noise, and the results here also suggest RR.

## 1. Introduction

In many engineering systems, noise disrupts the signals sent and received, making it impossible to maintain a correct input/output environment, so it is essential to reduce noise from the computational environment as much as possible. In contrast, living systems live in environments full of noise, and it is almost impossible to avoid noise. Therefore, many living systems, on the other hand, make good use of noise to improve various computational capabilities [1,2].

A well-known phenomenon that actively uses noise is stochastic resonance [3,4]. It is known that paddle-fishes use noise to enhance the weak electrical signals emitted by planktonic prey to detect them, and this is a typical example of stochastic resonance [5]. A similar example to stochastic resonance is recurrence resonance [6,7]. This does not enhance signals from the outside, but rather enhances the behaviour of attractors hidden within the systems, and strengthens the system’s potential through noise.

Notably, a method has been proposed that makes it easy to find recurrence resonance by increasing the noise level in a specific noise region, where the global mutual information increases, reaches a maximum value, and then decreases; the importance of recurrence resonance has been further increasing [7]. In this case, it is said that recurrence resonance easily appears if the connections between neurons are dense.

Recurrence resonance could be related to various dynamical systems that utilize internal noise. First, in neural networks that are the basis for deep learning, recurrent neurons [8,9] are equivalent to a ’heat bath’ in reservoir computing [10,11,12], and it is known that their role of stirring up inputs with noise is important. They play a role in moving between attractors generated internally by internal noise, and in that sense, are closely related to recurrence resonance.

Second, it has become clear that the distribution of avalanche firings in neurons follows a power law, and in recent years, discussions of self-organized criticality (SOC) [13,14,15] and edge of chaos [16,17,18] have rekindled in the experimental field of neuroscience [19,20,21]. These are understood as critical phenomena in the phase transition from order to chaos, and it is thought that the balance between the attractor of structure formation and noise (chaos) leads to efficient computation and a power law. However, SOC and edge of chaos require tuning to the critical value, and the generality of tuning even in SOC has not been clarified.

Another scenario for SOC and edge of chaos merits consideration. It involves a system in which order and chaos (noise) are well adjusted by using a synchronous random update of the system elements [2]. When this system is implemented with elementary cellular automata (ECA) [22,23], various power law behaviors can be realized without tuning, and in terms of computational power, while there is a strict trade-off between computational universality and computational efficiency in synchronous ECA, this is broken in asynchronous ECA, and it is known that both computational universality and efficiency are realized [2]. It has also been found that when such an asynchronously adjusted ECA is implemented in a neural network reservoir, learning performance is improved compared to using an ECA (class III) that behaves chaotically in the reservoir [24].

Furthermore, instead of providing asynchronous updating, by implementing an input–output relation which leads to quantum logic in a cellular automaton (quantum automaton) [25], it is easy to obtain a time evolution pattern that features both chaos and order. It has been found in recent years that a power law can also be realized by 1/f noise [26].

When considering the relationship between recurrence resonance, SOC, and quantum automata, the following problems have emerged. First, while recurrence resonance is judged by global mutual information, it is impossible to observe all neurons in the brain. Therefore, it is possible that recurrence resonance occurring as a whole can also be observed in parts, depending on the connection state of neurons. Mutual information has a small value in simple chaos or simple order, and its value is large only in complex patterns that have both characteristics. In other words, when recurrence resonance is observed in parts, it is thought that the parts behave like SOC. Second, therefore, in a system in which recurrence resonance can be judged not only in the whole network but also in the partial network, critical phenomenon-like behavior and even power law behavior may be realized. This can be judged by whether 1/f noise is found in the behavior time series of a single neuron in the neural network. Third, a network that realizes the first and second behaviors could have a partially sparse distribution, like a quantum logical automaton, where the connection state of neurons brings about quantum logic. The purpose of this study is to measure and evaluate the above in a Boltzmann machine [27,28].

The key point is the following. It was previously reported that RR can be easily realized if the connectivity is dense to some extent. In this paper, we propose the concept of local observability of RR and show that such observability depends on the distribution of connections among elements, giving rise to quantum logic.

In addition to neural networks, clusters of proteinoid microspheres are sparsely connected systems that emit signals like neural pulses and are regarded as one of various hopeful candidates for the system of the origin of life [29,30,31,32]. These are not freely connected to distant elements like neurons, but are clusters that depend on spatial structure. A cluster structure that has sparse parts while connecting neighbours in a spatially dependent manner can be considered a system that can be approximated to a Boltzmann machine that depends on spatial structure. Therefore, we inserted electrodes into part of the proteinoid microsphere cluster, observed the time series of the potential signal, and verified whether 1/f noise was observed by FFT. If this is possible, it would be a strong concrete example that supports our view.

## 2. Materials and Methods

### 2.1. Boltzmann Machine and Data Analysis

Recurrence resonance is here estimated in a neural network, Boltzmann machine [7]. A one-dimensional Boltzmann machine is expressed as(1)$$u_{i}^{t}=\sum_{j=1}^{N}w_{ij}s_{j}^{t}+r\eta_{i}^{t},$$
which involves a recurrent neuron and where(2)$$s_{i}^{t+1}=\tanh u_{i}^{t}.$$
The symbol $w_{ij}$ represents the weight in the connection, where $w_{ij}\in \left\{-W,0,+W\right\}$ and $w_{ij}∼N\left(0,1\right)$. Crucially, *W* is the weight strength, *r* is the noise strength, and *N* represents the system size. The $\eta_{i}^{t}∼N\left(0,1\right)$ are random numbers, drawn from a normal distribution with mean 0 and standard deviation 1. The time development of a one-dimensional network is expressed as(3)$$\left(s_{1}^{t},s_{2}^{t},\ldots ,s_{N}^{t}\right)\mapsto \left(s_{1}^{t+1},s_{2}^{t+1},\ldots ,s_{N}^{t+1}\right).$$
To consider a network dependent on spatial structure, we also estimate a two-dimensional Boltzmann machine expressed as(4)$$u_{ij}^{t}=\sum_{p=1}^{N}\sum_{q=1}^{N}w_{ijpq}s_{pq}^{t}+r\eta_{ij}^{t},$$
where(5)$$s_{ij}^{t+1}=\tanh u_{ij}^{t}.$$
For a two-dimensional network, NN represents the size of the system. The time development of a two-dimensional network is expressed as(6)$$\left(s_{11}^{t},s_{12}^{t},\ldots ,s_{NN}^{t}\right)\mapsto \left(s_{11}^{t+1},s_{12}^{t+1},\ldots ,s_{NN}^{t+1}\right).$$

The metric entropy and mutual information are calculated for a time series of neurons to evaluate the recurrence resonance. The metric entropy is defined by(7)$$H\left(X\right)=-\sum_{X}P\left(X\right)\log_{2}P\left(X\right),$$
where *X* is a binary sequence. Mutual information is defined by(8)$$I\left(X;Y\right)=\sum_{X}\sum_{Y}P\left(X,Y\right)\log_{2}\left(\frac{P(X,Y)}{P(X)P(Y)}\right),$$
where both *X* and *Y* are binary operations. If a binary sequence is defined for an array of all neurons such as(9)$$X=(s_{1}^{t},s_{2}^{t},\ldots ,s_{N}^{t})$$
for a one-dimensional network and(10)$$X=(s_{11}^{t},s_{12}^{t},\ldots ,s_{NN}^{t})$$
for a two-dimensional network, the entropy and mutual information are called global entropy and global mutual information. If a binary sequence is defined to have a small part of neurons, we add the term “local”, and they are called local entropy and local mutual information. In global mutual information, *Y* is defined as(11)$$Y=(s_{1}^{t+1},s_{2}^{t+1},\ldots ,s_{N}^{t+1})$$
for a one-dimensional network and as(12)$$Y=(s_{11}^{t+1},s_{12}^{t+1},\ldots ,s_{NN}^{t+1}).$$
A binary sequence of *Y* for local mutual information is also defined for a small part of a network.

We estimate the behavior of a neuron in terms of the power spectrum. Given a time series of one neuron, $u_{i}^{t}$, the power spectrum of a time series is defined as follows. First, the discrete Fourier transform for a time series is obtained by(13)$$C\left(f\right)=\frac{1}{T}\sum_{t=0}^{T-1}s_{i}^{t}e^{-i(2\pi tf)/T}$$
where f=0,1,…,T−1 is the frequency corresponding to the period, T/f. The power S(f) is obtained as the squared absolute value of the discrete Fourier transform such that(14)$$S\left(f\right)={\left|C\left(f\right)\right|}^{2}.$$

### 2.2. Logical Structure of Neuron Connections

In this paper, we investigate the effect of the distribution of weight of connection, which reveals Boolean and/or quantum logic in terms of lattice theory [33]. A lattice is a kind of algebraic structure, which is a partially ordered set closed with respect to binary operations, join, and meet. Using a rough set lattice technique, the binary relation can be interpreted as a lattice structure [34,35].

Figure 1 shows how a binary relation is interpreted as a lattice. A binary relation is set for a distribution of the weight of connection or joint probabilities of states of neurons. Define sequences of neurons $U_{1}=\left\{A,B,C,D,E\right\}$ and $U_{2}=\left\{a,b,c,d,e\right\}$ such that $\left\{s_{1}^{t},s_{2}^{t},s_{3}^{t},s_{4}^{t},s_{5}^{t}\right\}$ with N=5. A binary relation *I*⊆$U_{1}$×$U_{2}$ is obtained dependent on whether $w_{ij}$ exceeds a threshold value or not, and is expressed as(15)$$\left(s_{i}^{t},s_{j}^{t}\right)\in I\Leftrightarrow w_{ij}\geq \theta$$(16)$$\left(s_{i}^{t},s_{j}^{t}\right)\notin I\Leftrightarrow w_{ij}<\theta$$
In Figure 1, if $(s_{i}^{t},s_{j}^{t})\in I$, the corresponding cell is painted blue; otherwise, it is blank.

The closure operation is defined by the following. For any subset *X* of $U_{1}$, closure operation is expressed as(17)$$Cl\left(X\right)=R_{*}\left(r^{*}\left(X\right)\right),$$(18)$$r^{*}\left(X\right)=\left\{q\in U_{2}|pIq,p\in X\right\},$$
where pIq represents (p,q)∈*I*. For any subset of $U_{2}$, *Y*, $R_{*}\left(Y\right)$ is expressed as(19)$$R_{*}\left(Y\right)=U_{1}-\left\{p\in U_{1}|pIq,q\notin Y\right\}.$$
It is straightforwardly verified that a collection of fixed points of closure operations is a rough set lattice such that(20)$$L=\{X\subseteq U_{1}|Cl\left(X\right)=X\}.$$
As each element of a rough set lattice is a set, the order relation between elements is the inclusion relation.

As shown in Figure 1, a diagonal relation for which all diagonal elements are in the relation and any other elements are not in the relation reveals a Boolean algebra, and a relation for which there are some diagonal sub-relations surrounded by related cells (p,q) (i.e., pIq) reveals a non-distributive orthomodular lattice that corresponds to quantum logic. For instance, Cl({A})={A}, Cl({B,C})={B,C}, and Cl({C,D})=U. Thus, {C,D} is not a fixed point of closure operation. A Hasse diagram, finally obtained from a given binary relation, is expressed as a disjoint union of $2^{2}$- and $2^{3}$-Boolean algebras whose greatest and least elements are common [36].

### 2.3. Proteinoid Microsphere Experiments

Local points from clusters of proteinoid microspheres emit a pulse-like signal in a way that is similar to neurons.

#### 2.3.1. Preparation of Proteinoid Microspheres

The proteinoid synthesis used analytical grade L-glutamic acid (L-Glu, CAS: 56-86-0) and L-phenylalanine (L-Phe, CAS: 63-91-2) from Sigma Aldrich Ltd., Gillingham, UK (purity ≥ 98%). Proteinoid solutions were prepared using either deionized water or KNO_3_ (0.065 M) ionic solution. To synthesize L-Glu:L-Phe proteinoids, we mixed 2.5 g each of L-glutamic acid and L-phenylalanine. We then heated the mixture to its melting point in a reflux apparatus, with continuous agitation, until a homogeneous slurry formed. The molten mixture was subsequently cooled to 80 °C, diluted with deionized water, and stirred for 3 h. We isolated the resulting precipitate via vacuum filtration. Then, we obtained the final purified proteinoid powder through lyophilization [37].

#### 2.3.2. Bioelectric Signal Measurement Setup

The apparatus for measuring bioelectric signals from the proteinoid system used dual iridium–platinum electrodes (0.1 mm diameter). They were positioned 10 mm apart (Figure 2). We recorded bioelectric potential differences using an ADC-24 PicoLog system. We chose this system for its high-precision voltage measurement. The setup had an electrical stimulator for precise sample excitation. It also had a system to control the environment and maintain experimental conditions. This data acquisition system enabled real-time monitoring and analysis of bioelectric signals. It detected changes in proteinoid microsphere behaviour with precision under controlled stimulation.

#### 2.3.3. Morphological Characterization of Proteinoid Microspheres

We examined the proteinoids with an FEI Quanta 650 SEM. See Figure 3. The micrograph shows spherical structures, which are 2 to 3 μm in diameter. They have a smooth surface. The microspheres tend to cluster, but they stay intact. The image was made with a low field detector (LFD). It used a 7.50 kV voltage, a 5.0 mm working distance, and a 9.82 × 10^−1^ Torr chamber pressure. The samples were imaged without metallic coating under low vacuum (HFW: 11.0 μm). This enabled observation of the microspheres at 18,778× magnification in their native, non-conductive state, which provided a clear view of their surface and spatial organization.

## 3. Results

### 3.1. One Dimensional Network

First, we describe the behavior of a Boltzmann machine with five neurons arranged in a row. We set various connections between neurons.

Figure 4 shows the global entropy and global mutual information of a specific Boltzmann machine that consists of five neurons plotted against the noise strength, where the global entropy shows a monotonic increasing function and the other curve of shows the global mutual information. The weight of the connection is determined subject to a normal distribution, which is called the normal condition. The color of the curves show the weight strength. The probability of a binary sequence P(X) is obtained by the frequency of the binary sequence in the time development of 10,000 steps, where $X=(s_{1}^{t},s_{2}^{t},s_{3}^{t},s_{4}^{t},s_{5}^{t})$.

It is clear that the global mutual information increases in the low range of the noise strength. The range in which the global mutual information increases suggests some explicit behaviour based on attractors in the neural network under the normal condition, which indicates recurrence resonance. As each neuron mutually interacts with each other under the normal condition in which each neuron is basically connected to all neurons in the network, the effect of the external noise is weakened by the interaction and is prevailed slowly. Therefore, the global entropy slowly increases as the weight strength increases—W=10. The global mutual information also slowly increases in the smaller range of the noise strength for the high weight strength. It is easy to see that the global mutual information gradually increases and reaches the maximal point till the global entropy reaches about the value of 3.6 independent of the connection strength. The global entropy reaches 3.6 for the noise strength of 1.3 if the weight strength is 1, and it reaches 3.6 for the noise strength of 10.0 if the weight strength is 10. This implies that the recurrence resonance is explicit if the weight strength is high.

Figure 5 also shows global entropy and global mutual information pairs of a specific Boltzmann machine which consists of five neurons, plotted against the noise strength, where the weight of the connection is determined under the diagonal distribution (left) and under the quantum distribution (right)—these are called the diagonal condition and Quantum condition, respectively. The diagonal distribution of the connection implies that the weight of a connection such as $w_{ii}$ is stochastically determined in the range between 0.7 and 1.0, and any other weight such as $w_{ij}$ (i≠j) is stochastically determined in the range of 0.0 and 0.05. The quantum distribution of the connection implies that weight of a connection such as $w_{ii}$ is stochastically determined in the range between 0.7 and 1.0, each of $w_{14}$, $w_{15}$, $w_{24}$, $w_{25}$, $w_{34}$, $w_{35}$, $w_{41}$, $w_{42}$, $w_{43}$, $w_{51}$, $w_{52}$, and $w_{53}$ is stochastically determined in the range between 0.2 and 0.3, and any other weight is stochastically determined in the range of 0.0 and 0.05. Global entropy and global mutual information pairs were obtained from a time series of 1000 steps (above) and 10,000 steps (below).

The global entropy under the diagonal condition rapidly increases even for the small noise strength, as the connection between neurons is so weak that each neuron behaves independently of other neurons; the effect of external noise is not weakened due to the interactions between neurons. While the global entropy increases until it reaches about 4.0, the global mutual information also increases until it reaches the maximal point. This tendency is found for both cases in which the probability of binary sequence was obtained in time series of 1000 steps and 10,000 steps. Although the range showing the increase in global mutual information is very narrow, the maximal value of global mutual information under diagonal conditions is very high. This shows that the spatial and temporal patterns under the diagonal condition are very complex between order and chaos.

Compared to other conditions, in the diagonal condition, the maximum value of global mutual information increases rapidly as the weight strength increases. In the normal condition in Figure 4 and the quantum condition in Figure 5, the maximum value of global mutual information hardly changes when the weight strength is changed from 1 to 2, to 5 and then 10. However, in the diagonal condition, when the weight strength is changed from 1 to 10, the maximum value of global mutual information changes about three times. In the diagonal condition, there is no interaction with other neurons, but as the weight strength increases, the autocatalytic effect becomes stronger, which is thought to enhance the tendency of recurrence resonance.

Compared to the behaviour of the Boltzmann machine under the diagonal condition, the global entropy and global mutual information under the quantum condition slowly increase, since the interaction among neurons can contribute to weakening the effect of external noise. The global mutual information increases and reaches the maximal point until the global entropy increases and reaches the value of 3.5, whether the probability was obtained in a time series of 1000 steps or 10,000 steps. The maximal value of the global mutual information is rather small compared to the diagonal condition.

If the probabilities of binary sequences are obtained in a time series of 1000 steps, the global mutual information converges not to 0.0 but to 1.0, regardless of whether the connection of neurons is subject to the diagonal or quantum condition. However, if the probability is obtained in a time series of 10,000 steps, it converges to about 0.0. This implies that if the time development proceeds for long enough time, then explicit complex patterns disappear because of the effect of external noise, and that recurrence resonance appears in the transient stage before the behavior of neural nets converges to a noisy steady state.

Figure 6 also shows global entropy and global mutual information against the noise strength of a Boltzmann machine consisting of seven neurons, where $X=(s_{1}^{t},s_{2}^{t},s_{3}^{t},s_{4}^{t},s_{5}^{t},s_{6}^{t},s_{7}^{t})$ for P(X). The probability of the binary sequence was obtained from a time series calculated over 10,000 time steps. It is easy to see that the patterns of the weight of the connection reveal Boolean algebra and quantum logic. As mentioned in the above section, the diagonal relation reveals Boolean logic. As the pattern of weight in the diagonal condition leads to the diagonal relation by the relation to which only the weight exceeding 0.7 belongs, the weight pattern in diagonal condition reveals Boolean algebra. The pattern of the weight in the quantum condition leads to a relation in which 3 × 3 and 4 × 4 diagonal sub-relations are surrounded by related cells, if the weight exceeding 0.2 is determined as an element of the relation; otherwise, the weight is determined as no relation. This pattern leads to quantum logic, which is expressed as a disjoint union of $2^{3}$- and $2^{4}$-Boolean algebras whose greatest and least elements are common. We also calculated global entropy and mutual information for the weight pattern, which revealed a disjoint union of two $2^{2}$- and one $2^{3}$-Boolean algebras under the quantum condition. However, they are omitted because the tendency of the entropy and mutual information is almost same as that in the graph with the quantum condition in Figure 6.

The trends of global entropy and global mutual information do not change fundamentally even when the number of neurons is increased. In the normal condition and quantum condition, the increase in global entropy and global mutual information in response to noise strength is gentle, and the maximum value of global mutual information hardly changes with changes in connection strength. In contrast, in the diagonal condition, both global entropy and global mutual information rise rapidly in response to an increase in noise strength, and as the connection strength increases, the maximum value of global mutual information also increases accordingly. This is thought to be because, as mentioned above, the interaction between neurons is weak, and the autocatalytic effect due to the connection strength is accentuated by the increase in connection strength.

Although the number of calculation steps is 10,000, because the number of neurons is increasing, the state does not converge to a sufficiently stable steady state. Therefore, the entropy and mutual information curves are not stable in the normal condition, and the mutual information converges to 1.0 instead of 0.0 in the diagonal condition and quantum condition.

### 3.2. Local Estimation in Two-Dimensional Network

It was reported that recurrence resonance could be observed not only in Boltzmann machines but also in various networks such as Hopfield [38] networks if the connections between neurons are dense [7]. This is certainly true. However, the way in which it manifests itself clearly differs depending on the pattern of connections between neurons. As we have seen so far, when the connections are dense as in the normal condition, the interactions between neurons effectively adjust the external noise, emphasizing the behaviour of the attractor of the network itself; while showing recurrence resonance and at the same time mitigating the effects of the noise, the system slowly reaches a steady state. In contrast, in the diagonal condition, recurrence resonance is also shown, but the effects of noise appear directly and do not seem to be significantly mitigated. In the quantum condition, the connection distribution seems to satisfy both characteristics, and recurrence resonance is shown while the effects of noise are also mitigated. Here, there is a struggle between the order latent inside the network and the noise (chaos) from the outside, and it is an interesting problem how the two are mediated in spacetime.

First, we consider a 3 × 3 neural network to consider the information-theoretic relationship between the whole and the part. The condition of connection between neurons is the normal condition. Global entropy and mutual information classify the nine-bit states by looking at all nine neurons and calculating the probability. In other words, the states are 000000000, 000000001, …, 111111111, and the probability is calculated for these $2^{9}$ states. In contrast, local entropy and mutual information are calculated from only three neurons. In this case, the states are 000, 001, 010, 011, 100, 101, 110, and 111, and the number of states is only $2^{3}$. Also, local entropy and mutual information are calculated only for three neurons, $s_{11}^{t}$, $s_{12}^{t}$, and $s_{13}^{t}$, rather than calculating the probability for every three bits for all neurons.

Figure 7A shows the global entropy and mutual information plotted against noise strength for a 3 × 3 neural network. As shown in the graphs of Figure 4, Figure 5 and Figure 6, the different colours represent different connection strengths. The probability is calculated over a time evolution of 50,000 steps. Because of the large number of neurons, as the connection strength increases, the interaction weakens the noise effect, and the time to fall into a noisy steady state becomes long enough, and the curve becomes more uneven. However, the important point is that in the low-noise strength region, there is a clear increase in mutual information and the local maximum. In other words, the global mutual information shows the existence of recurrence resonance. However, the local mutual information does not show the existence of recurrence resonance (Figure 7B). It only decreases monotonically with increasing noise intensity. There is no region where the value increases, which means that no complex pattern appears in any noise region.

Although a whole system realizes recurrence resonance by mitigating noise, such a tendency is not observed when only a part of the system is examined. The behaviour of the Boltzmann machine in the normal condition means that the conflict between chaos and order is not observed locally. Mutual information reduces the value of both chaotic patterns and regular patterns such as periodic oscillations; so, locally, there is either chaos or order, and neither of which are mixed in a complex pattern.

However, Figure 8 shows that this is not a general tendency. It shows the behaviour of local entropy and local mutual information in a Boltzmann machine network consisting of 25 neurons arranged two-dimensionally in a 5 × 5 grid. The horizontal axis is the noise strength, and the colour of the curve indicates the weight strength, as in Figure 4, Figure 5, Figure 6 and Figure 7. Also, all the monotonically increasing curves are local entropy, and the other pairs that do not are mutual information, as shown in Figure 4, Figure 5, Figure 6 and Figure 7. The probability is calculated from the frequency counted over 10,000 steps, with the state defined as $X=(s_{11}^{t},s_{12}^{t},\ldots ,s_{15}^{t})$.

The 5 × 5 circle distribution shown in each graph in Figure 8 does not directly show the weight distribution of the connections. The circles represent neurons, and in the normal condition, the connection weight between all neurons is determined by probabilities that follow a normal distribution. In the blocky, diagonal, and quantum conditions, the connection weight between dark blue neurons is determined between 0.7 and 1.0, and between white neurons, the connection weight is determined between 0.0 and 0.05. In the quantum condition, the connection weight between light blue neurons is determined between 0.2 and 0.3. The difference is clear when comparing the normal condition with the other conditions—blocky, diagonal, and quantum. In all conditions except for the normal condition, an increase in value and a maximum value are observed in the local mutual information at low noise strength. In other words, under these conditions, recurrence resonance is clearly observed, and a complex pattern that has both chaos and order appears locally. This is thought to be due to the fact that the coupling has sparse parts, which is different from the normal condition.

### 3.3. 1/f Noise in Boolean and Quantum Condition

In the previous section, it was shown that a mixed state of chaos and order can be observed even in the spatial part, depending on the connection state of the neuron. This indicates a phase-transitional critical state in the phase transition from order to chaos. So, can a phase-transitional critical state be found in a single neuron? To evaluate this, a single neuron was selected from the network, and its time series of states was analyzed by Fourier transformation. The time series was calculated from 50 series with random initial values, and the power spectrum was averaged.

Figure 9 shows the power spectrum plotted against frequency in a log–log plot. The network of this Boltzmann machine is composed of five neurons, and the 5 × 5 square of circles shows the weights of the neuronal connections. The light orange circle in the normal condition indicates that all weights follow a normal distribution. In the blocky, diagonal, and quantum conditions, the dark orange circles show weights between 0.7 and 1.0, the white circles show weights between 0.0 and 0.05, and the light orange circles show weights between 0.2 and 0.3. The power spectrum in the normal condition clearly shows white noise, and no critical behaviour is observed. In the blocky condition, the slope is a little bit larger than 1.0, but there is a tendency toward a power law distribution. Note that the blocky condition also reveals Boolean algebra with respect to rough set lattice transformation. In the diagonal and quantum conditions, there is clear 1/*f* noise in the high-frequency components. This means that the phase transition critical phenomenon is observed in these connection patterns.

Recall the 5 × 5 = 25 network (Figure 8), which also shows recurrence resonance in local mutual information. Here, we again selected one neuron, analyzed its time series by fast Fourier transformation with different initial values, and plotted the power spectrum averaged over the time series of 50 trials against frequency, and displayed the graph in a log–log plot (Figure 10). Here, again, the tendency of the power spectrum depending on the neuron’s connection type is almost the same as in Figure 9. In the normal condition, it shows white noise, and in the blocky condition, it shows a tendency toward a power law distribution with a slope of −1.0. In the diagonal and quantum conditions, it shows what is called an exact 1/f noise with a slope of −1.0. However, unlike the case of the 5-network, it shows 1/f noise for low-frequency components.

In the quantum condition and diagonal conditions, the behavior is extremely similar, but since 1/f fluctuations are shown for low-frequency components, we can say that there is a clear critical behavior. The distribution pattern of the connection strength between neurons clearly plays a role, and it is thought that the distribution in which there are areas of strong connection and areas of sparse connection is important. In the quantum condition, the connection strength of the non-diagonal sub-relation area (corresponding to the area of strong connection in the blocky condition) is set between 0.2 and 0.3, but it is possible that a difference from the diagonal condition was not observed because this value is relatively small. In the quantum condition, the transition between the neurons arranged diagonally and the area of strong connection in the blocky condition is cut off, so when the transition between these areas is allowed, the frequent transition between chaos and order is further enhanced, and 1/f noise may be observed not only in low-frequency components but also in all frequency components. In any case, the increase in local mutual information observed at low noise intensity and the existence of a maximum value not only show the criticality of chaos and order in the subspace within the system, but also in the temporal changes of a single neuron.

Although Figure 8 shows that only the low-frequency components show the power law, such a property is influenced by the connection strength in the non-diagonal block part. Figure 11 shows the power spectrum for a time series of the state of a neuron plotted against the frequency, where the connection strength in the block part changes in various ways. If the connection strength is set between 0.25 and 0.65 (Figure 11a), the power law is noticeable in the high- and low-frequency components.

So, are these connection strengths (and so on) determined arbitrarily? In excess Bayesian estimation and inverse Bayesian estimation [39], which apply Bayesian inference locally to the connection probability distribution, it is known that the connection probability distribution becomes a distribution similar to the quantum condition. In that case, the connection probability, which corresponds to the connection strength, is determined autonomously by Bayesian and inverse Bayesian inference. It is highly likely that such a mechanism is involved in 1/*f* noise, and further investigation will be required.

Local recurrence resonance observability (LRRO), for which RR can be detected only from locally parts of a neural network, could be a condition that the assemblage has a blank area and shows a cluster-like structure. Conversely, if such a material situation is allowed, LRRO can be observed regardless of the form of interaction or the form of the dynamical system, and it is possible that the characteristics of the critical phenomenon such as 1/*f* fluctuations associated with it can be found. Forms such as Boltzmann machines are usually considered to be advanced dynamical systems that implement majority voting, such as highly organized neuronal systems. If the critical phenomenon originates from a dynamical system, it would be misguided to compare the Boltzmann machines with the proteinoid microsphere. However, the LRRO found here and the critical phenomenon caused by it may be a characteristic that depends on the characteristics of the execution environment that executes computation and the distribution pattern of the element connections, rather than the characteristics of the dynamical system. Therefore, although the connection between the two is conceptual, we decided to investigate the membrane potential time series of proteinoid microspheres with cluster-like connections and focus our experiments on whether or not the 1/*f* fluctuations seen in LRRO can be observed.

### 3.4. 1/f Noise in Proteinoid Microsphere

As mentioned before, the proteinoid microsphere is generated by polymerization of aminoacids, and constitutes a cluster structure of microspheres. A membrane potential maintains a pulse-like signal as well as a spike emitted from a neuron [40]. We here show that a time series of membrane potential reveals a typical 1/f noise, as shown in Figure 12.

Figure 12 shows a time series analysis and FFT for all eight channels. Each channel shows typical fluctuations in membrane potential (V(t)) and their power spectral density (|P(f)|). The power spectra follow a power law relationship:(21)$$\left|P\left(f\right)\right|=Cf^{-\beta }$$
where (*f*) represents frequency, (*β*) is the scaling exponent, and (*C*) is a constant. Taking the logarithm of both sides yields the following:(22)log|P(f)|=−βlog(f)+log(C)
The power spectra analysis shows scaling exponents from −0.43 to −1.07. This confirms (1/f)-like noise in all channels. The membrane potential time series (V(t)) can be characterized by its autocorrelation function:(23)R(τ)=〈V(t)V(t+τ)〉
where (τ) is the time lag and (〈·〉) denotes time averaging. In our analysis, we define two distinct frequency regimes: the low-frequency regime ($f\leq 10^{-2}$ Hz, corresponding to timescales longer than 100 s) and the high-frequency regime ($f>10^{-2}$ Hz, corresponding to timescales shorter than 100 s). The power spectrum scaling behaviour is characterized by $P\left(f\right)∼f^{\beta_{low}}$ for low frequencies and $P\left(f\right)∼f^{\beta_{high}}$ for high frequencies, where $\beta_{low}$ ranges from −1.30 to −0.51 and $\beta_{high}$ ranges from −1.07 to −0.41 across different channels. This division reveals different scaling behaviours at different temporal scales, with the low-frequency regime generally capturing long-term system dynamics and the high-frequency regime reflecting faster fluctuations. The Wiener–Khinchin theorem [41] links the power spectral density to the autocorrelation function.(24)$$\left|P\left(f\right)\right|=\int_{-\infty }^{\infty }R\left(\tau \right)e^{-2\pi if\tau }d\tau$$
The amplitudes across channels vary by two orders of magnitude. They range from about 2 mV in channel 8 to 200 mV in channel 2. This shows a significant unevenness in the microsphere distribution and the relevant electrical activity.

The root mean square fluctuation (*F*(Δt)) of the voltage signal over a time window (Δt) scales as follows:(25)$$F\left(\Delta t\right)\propto \Delta t^{\alpha }$$
where ($\alpha_{theo}$) is related to the power spectral exponent (β) by ($\alpha_{theo}=\left(1+|\beta |\right)/2$). The analysis of bioelectric signals from proteinoid microspheres reveals complex dynamics with distinct behaviour in low- and high-frequency regimes, as summarized in Table 1. Several channels show strong long-range correlations in the low-frequency regime, particularly channels 2, 3, and 4 ($|\beta_{low}|\;>1$). The theoretical scaling exponents ($\alpha_{theo}$) range from 0.705 to 1.15, with the low-frequency regime generally showing higher values (0.755–1.15) compared to the high-frequency regime (0.705–1.035). Notably, the directly fitted scaling exponents ($\alpha_{fit}$) show systematically lower values than the theoretical predictions, ranging from 0.32 to 0.90, suggesting a more complex relationship between temporal correlations at different scales. The high-frequency regime shows more consistent scaling behaviour across channels, particularly evident in channels 1, 2, 4, 5, and 7 ($\alpha_{fit,high}>0.80$). This dual-regime analysis reveals that the temporal organization of these bioelectric patterns combines both slow, strongly correlated fluctuations and faster, more varied dynamics, resembling the multi-scale characteristics observed in biological membrane potentials.

## 4. Discussion

Since Boltzmann machines are originally free to connect between neurons, a space-dependent expression such as a two-dimensional form has no specific meaning. However, in this paper, we consider not only neurons but also proteinoid microspheres to be primitive neurons that emit pulse-like waves, and we think that a phenomenon like recurrence resonance may be hidden in the cluster of proteinoid microspheres. This is why we assumed a space-dependent network that strongly connects between neighboring neurons and a two-dimensional form such as a 5x5 network. Here, we will discuss the local behavior of recurrence resonance and its significance, including the relationship with proteinoid microspheres.

In complex dynamical systems, even if a system starts from random initial values, there are attractors that are not excited from there. A typical example is the Garden of Eden in the Game of Life, which cannot be approached from other states [42]. How can we access such hidden attractors? One of the most hopeful possibilities is recurrence resonance (RR). RR exposes hidden attractors by using external fluctuations. However, if the noise is too large, the behavior of the attractor is drowned out by the noise, so the attractor appears only in a specific low-noise region. Naturally, a certain state of equilibrium between the chaos caused by noise and the order caused by the attractor will emerge here. This is thought to be exactly the critical state discussed in SOC and edge of chaos.

In this paper, we performed numerical calculations based on the expectation that RR and critical states are related. First, we evaluated how the appearance of RR is related to the connection strength pattern of neurons. RR is said to appear if there is a connection between neurons, but it was predicted that there is a difference in its behavior. If the connection strength is massive and follows a normal distribution, such complex behavior will not appear. This is because it is the local loss of information that creates the complex structure. RR can be evaluated by the increase in global mutual information and the existence of a maximum point against the noise strength, so we first calculated global mutual information against the noise strength for small networks consisting of five or seven neurons, where global mutual information can be calculated. The results showed that the maximum value was large and the noise region where global mutual information increased was wide in those regions with a connection strength pattern that indicated Boolean algebra (diagonal condition) or quantum logic (quantum condition) with a connection strength defect.

Secondly, the value of mutual information is low for both random and simple periodic oscillations (order), and is high for complex patterns that are critical states between chaos and order. So, how is this behavior realized as a complex pattern? If there is a local defect in the connection strength, will a complex pattern also appear locally? Therefore, we prepared 5 × 5 neural networks that strengthen the connection strength between nearby neurons depending on the space, and changed the connection strength to normal, diagonal, and quantum conditions to evaluate whether RR appears in local space. This was calculated and evaluated by local mutual information, which calculates mutual information only for a part of the network. No increase in local mutual information was observed in the normal distribution, but the increase and maximum value are shown in the diagonal and quantum conditions, and RR could be clearly recognized even locally.

In the case of the normal condition, the mutual information only decreases monotonically, which means that the pattern found in the initial value becomes a random pattern influenced by noise, and the value decreases, or a clear vibration pattern appears, and then the value decreases. In other words, no critical behavior with a mixture of chaos and noise was found locally. In contrast, the phenomenon found in the diagonal and quantum conditions, where the mutual information increases locally and reaches a maximum value, means that a critical phenomenon in which chaos and order are balanced appears locally. In other words, this means that critical behavior appears locally.

Therefore, we selected neurons from the network that were numerically calculated, analyzed the time series with FFT, and investigated whether the so-called 1/f noise appears. $f^{-\alpha }$ noise was clear in the diagonal and quantum conditions, where RR was found in the local mutual information. This 1/f noise was found in the high-frequency components of the one-dimensional array, and in the two-dimensional 5 × 5 array, it was found in the low-frequency components. This is because when there is a local defect in the connections of neurons, the flow of information is adjusted, the noise is suppressed at various periods, and the distribution is not such that there is an average value in the periods, resulting in the appearance of $f^{-\alpha }$ noise.

Is it possible that such RR can be observed in other than real neural networks? Proteinoid microspheres, which are materials associated with the origin of life, are known to emit pulse-waves like neurons when electrodes are inserted into their aggregations and their membrane potential is measured. Since the aggregations form a three-dimensional structure with gaps in the form of clusters, the two-dimensionally arranged Boltzmann machines that we have taken up can be considered to approximate this. Here, we measured the time series of potential activity in eight channels in a cluster of Glu-Phe proteinoid microspheres and analyzed the time series by FFT. Finally, we obtained the typical 1/f noise in which the power showed a power law distribution with respect to frequency. Because the environment of Glu-Phe proteinoid microspheres was exposed to noise in solution, this was thought to suggest RR. Whether the 1/f noise shown by Glu-Phe proteinoid microspheres truly shows RR or not will be clear if we change the noise strength and measure the local mutual information against it. In that case, measuring with eight channels would be sufficient to compute local mutual information.

In the future, we will need to give more thought to mechanisms that create critical behaviour through wandering between attractors and enhance computational power, rather than simply using noise to summon hidden attractors.

## 5. Conclusions

Originally, RR was calculated by global mutual information (GMI), which calculates the probability from the arrays of the states of all elements, and was defined by the increase in GMI under a specific region of noise strength. However, in a system with a large system size, this cannot be calculated in principle. Here, we found a case where RR can be found by local mutual information (LMI), which calculates the probability from only a part of a neural network. This is local RR observability (LRRO), which depends on the connection state of the neurons and appears in a connection distribution with a certain degree of lack of connection. In other words, LRRO does not exist in normal conditions, but LRRO can be found in diagonal conditions, blocky conditions, and quantum logic conditions. Furthermore, many of the cases where LRRO was found show characteristics of critical phenomena such as 1/f, and strict 1/f fluctuations can be found, especially in quantum conditions. Since LRRO is a partial increase in the local MI in a local neuron, it suggests that a mixed state of chaos and order is realized locally. MI shows high values in complex patterns that are not simply random or simply periodic oscillations. Therefore, an increase in MI should at least be a necessary condition for complex critical phenomena. In other words, there is no doubt that some LRROs exhibit critical phenomena. We therefore investigated the time series of a single neuron under the coupling conditions that show LRRO, namely, diagonal, blocky, and quantum logic conditions, and evaluated whether 1/f fluctuations, a characteristic of critical phenomena, could be observed. As a result, we were able to find 1/f fluctuations in these cases, and in particular, we found that strict 1/f fluctuations—1/f fluctuations with an exponent extremely close to 1.0—could be found under the quantum logic condition. This may serve as a different scenario for self-organized critical phenomena and edge of chaos. It was believed that the space of possible dynamical systems originally assumed was mostly either order or chaos, and that only a very small number of dynamical systems in between exhibited critical phenomena. From this, the reason why only a small number of dynamical systems are often found in biological systems was attributed to strict natural selection. However, it may not be the dynamical system itself, such as the mode of interaction, that causes criticality, but rather that criticality appears in more fundamental situations, such as the conditions and environment that realize the dynamical system. In quantum logic automata, various transition rules for time evolution are prepared, while in the execution environment that realizes them, deterministic ones and ones that allow ambiguity are prepared. The former is an execution environment that leads to classical logic, and the latter is an execution environment that leads to quantum logic. It was found that criticality is rarely observed in a classical logic environment, but in a quantum logic environment, 1/f fluctuations are observed in many cases regardless of the transition rules, and criticality is observed.

The criticality shown by the LRRO we found is also dependent on the bond distribution pattern, which strengthens the scenario of criticality depending on the computation environment. In other words, the reason for the criticality is not due to severe competition for survival, but is due to the distribution of materials that are originally widely present, and it may be something that can be found very easily. The so-called 1/*f* fluctuation was observed from the local time series of the proteinoid microspheres. Although it is not strictly 1/*f*, even in the case of the quantum logical condition of the Boltzmann machine, the exponent is different between the high-frequency component and the low-frequency component, so it is possible that 1/*f* fluctuation was observed in a broad sense. This makes it possible to think that even if there is a difference in the dynamical system between Boltzmann machines and proteinoid microspheres, there is a common condition for both to behave like a critical phenomenon, from which it can be inferred that this is the binding structure of the elements.

## Figures and Tables

**Figure 1 entropy-27-00145-f001:**
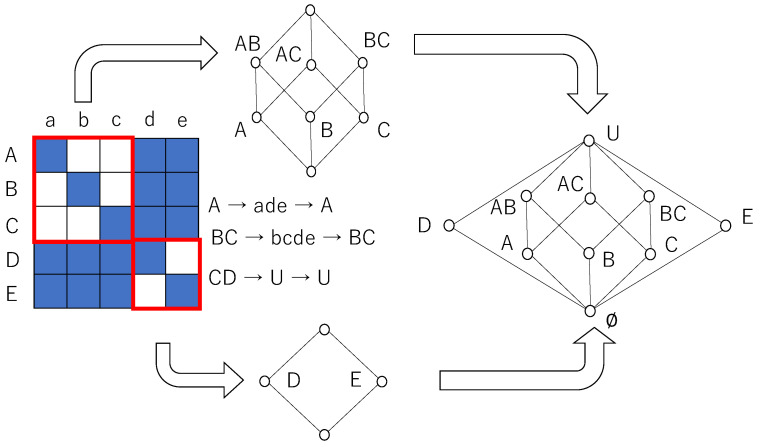
A lattice obtained from a binary relation by using a closure operation. If a relation is n×n diagonal relation, then its corresponding lattice is a $2^{n}$-Boolean lattice consisting of $2^{n}$ elements. A binary relation (left) consists of $2^{3}$- and $2^{2}$-diagonal sub-relations surrounded by related cells. As each $2^{n}$-sub-relation reveals a $2^{n}$-Boolean sub-algebra, a rough set lattice results in a disjoint union of $2^{3}$- and $2^{2}$-Boolean sub-algebras in which the least and greatest elements are common to both sub-algebras.

**Figure 2 entropy-27-00145-f002:**
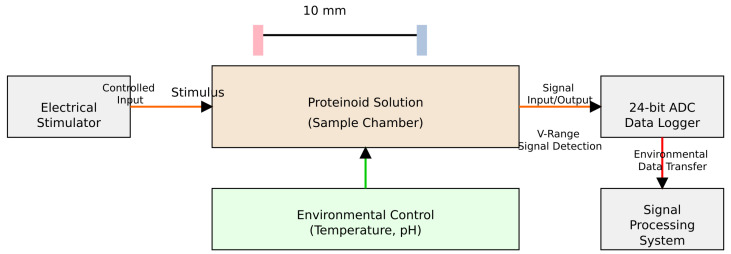
Experimental configuration for measuring bioelectric signals in the proteinoid micropsheres. The apparatus has a pair of iridium–platinum electrodes (0.1 mm diameter). They are 10 mm apart for detecting potential differences. A high-precision ADC-24 PicoLog data logging system performs signal acquisition. The setup has three key components: a precision electrical stimulator for controlled excitation; an environmental monitoring system for stable conditions; and a signal processing unit for real-time analysis and visualization.

**Figure 3 entropy-27-00145-f003:**
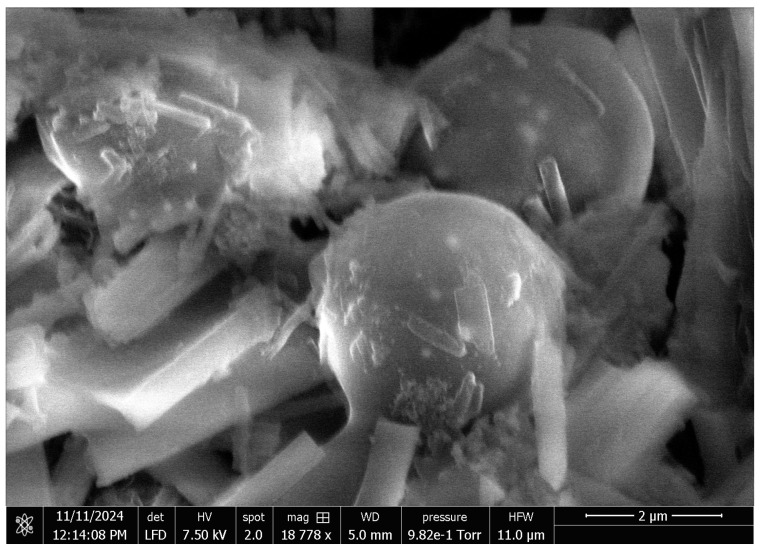
A scanning electron microscopy (SEM) image of uncoated proteinoid microspheres (L-Glu:L-Phe). It shows their spherical shape and distinct surface features. The image was acquired using a low field detector (LFD) at 7.50 kV accelerating voltage, with a working distance (WD) of 5.0 mm and chamber pressure of 9.82 × 10^−1^ Torr. Imaging the non-conductive sample without a metallic coat was possible by operating under low vacuum (HFW: 11.0 μm). The scale bar represents 2 μm at 18,778× magnification.

**Figure 4 entropy-27-00145-f004:**
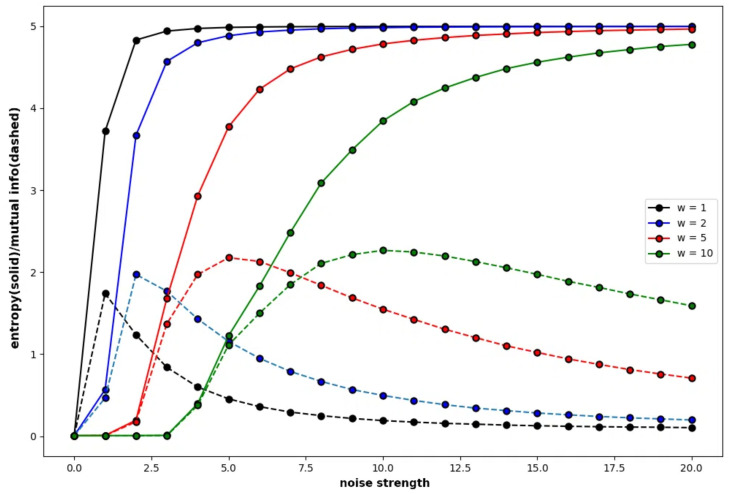
Four pairs of global entropy and global mutual information plotted against noise strength for the Boltzmann machine consisting of five neurons, under the normal condition. The weight of the connection between neurons is determined by the normal condition, where $w_{11}=-0.6$, $w_{12}=-2.0$, $w_{13}=-2.2$, $w_{14}=0.7$, $w_{15}=-1.1$, $w_{21}=-0.1$, $w_{22}=0.3$, $w_{23}=0.3$, $w_{24}=-1.2$, $w_{25}=-0.7$, $w_{31}=0.5$, $w_{32}=1.4$, $w_{33}=-0.6$, $w_{34}=-0.4$, $w_{35}=1.5$, $w_{41}=1.9$, $w_{42}=0.1$, $w_{43}=-0.6$, $w_{44}=-0.8$, $w_{45}=-1.2$, $w_{51}=-0.5$, $w_{52}=1.5$, $w_{53}=-1.0$, $w_{54}=-0.5$, and $w_{55}=-1.6$. The color of the curves represents the difference of the weight strength multiplied by the weight of connection.

**Figure 5 entropy-27-00145-f005:**
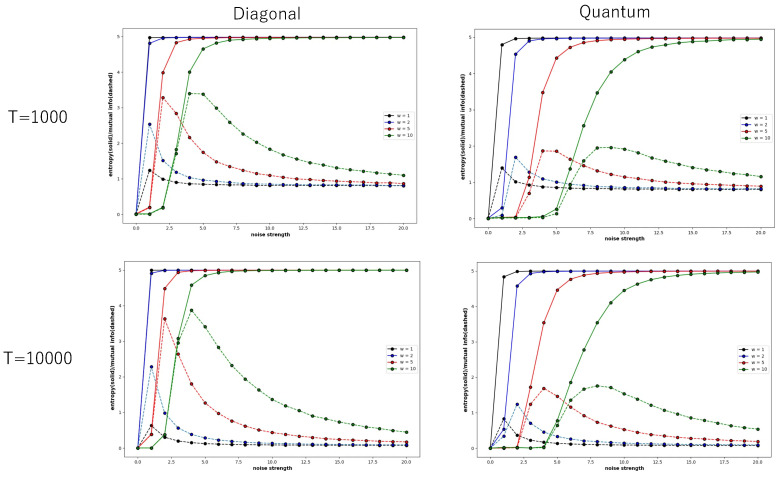
Four pairs of the global entropy and global mutual information plotted against noise strength for the Boltzmann machine consisting of five neurons, under the diagonal (left) and quantum conditions (right). The global entropy and mutual information were calculated from a time series of 1000 steps (above) and of 10,000 steps. The color of a pair of curves represents the weight strength, where black, blue, red, and green curves represent W=1, W=2, W=5, and W=10, respectively.

**Figure 6 entropy-27-00145-f006:**
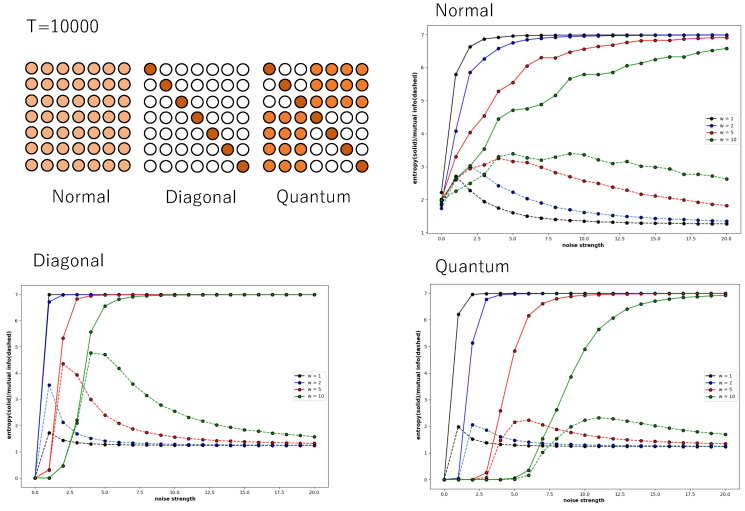
Four pairs of global entropy and global mutual information plotted against noise strength for a Boltzmann machine consisting of seven neurons under the normal, diagonal, and quantum conditions. The square consisting of 7 × 7 circles (above left) represents a set of weights of the connections between neurons, where dark brown circles arranged along the diagonal line represent the weight stochastically determined between 0.7 and 1.0, orange circles represent the weight stochastically determined between 0.2 and 0.3, and white circles represent the weight stochastically determined between 0.0 and 0.05. Pale orange circles represent the weight stochastically determined and subject to a normal distribution. The color of a pair of curves represents the weight strength, where black, blue, red, and green curves represent W=1, W=2, W=5, and W=10, respectively.

**Figure 7 entropy-27-00145-f007:**
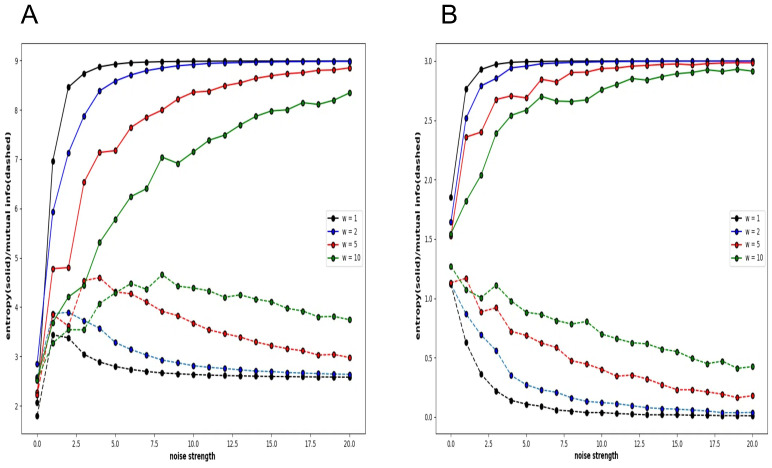
Four pairs of global and local entropy and mutual information plotted against noise strength for Boltzmann machine consisting of 3 × 3 neurons under normal condition. (**A**). Global entropy and global mutual information. (**B**). Local entropy and local mutual information.

**Figure 8 entropy-27-00145-f008:**
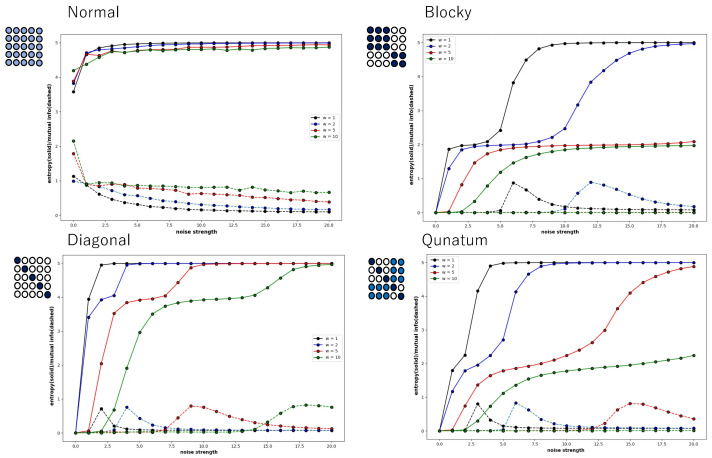
Four pairs of local entropy and local mutual information plotted against noise strength for Boltzmann machine consisting of 5 × 5 neurons, under normal, blocky, diagonal, and quantum conditions. Square consisting of 5 × 5 circles represents set of neurons suggesting weight of connections (see text). Colour of pair of curves represents weight strength, where black, blue, red, and green curves indicate W=1, W=2, W=5, and W=10, respectively.

**Figure 9 entropy-27-00145-f009:**
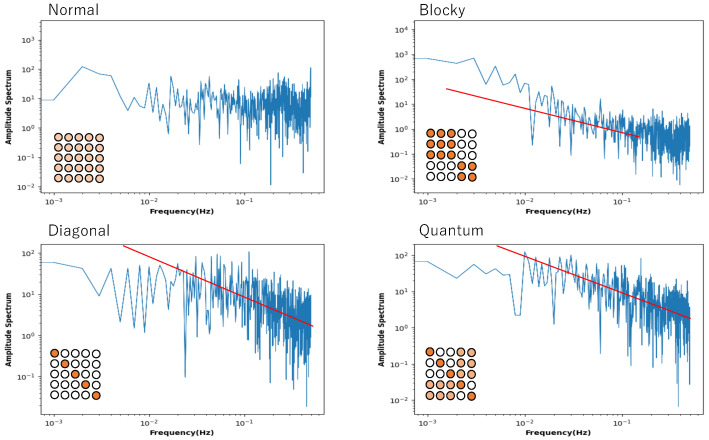
Power spectrum for a time series of the state of a neuron plotted against frequency in a log–log plot. The neuron was chosen from a Boltzmann machine network consisting of 5 neurons. The square consisting of 25 circles represents the distribution of weights of the connection of neurons. The red line in the diagonal and quantum conditions represents a slope of −1.0.

**Figure 10 entropy-27-00145-f010:**
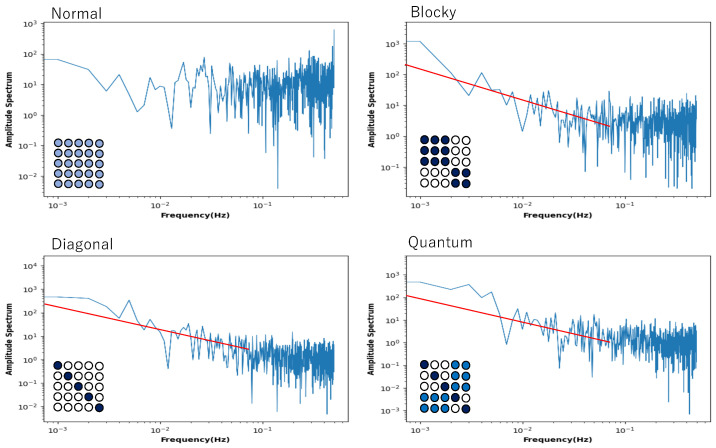
Power spectrum for a time series of the state of a neuron plotted against frequency in a log–log plot. The neuron was chosen from a Boltzmann machine network consisting of 5 × 5 = 25 neurons. The square consisting of 25 circles represents the distribution of neurons, revealing the connectivity between neurons by color. The red line in the diagonal and quantum conditions represents a slope of −1.0.

**Figure 11 entropy-27-00145-f011:**
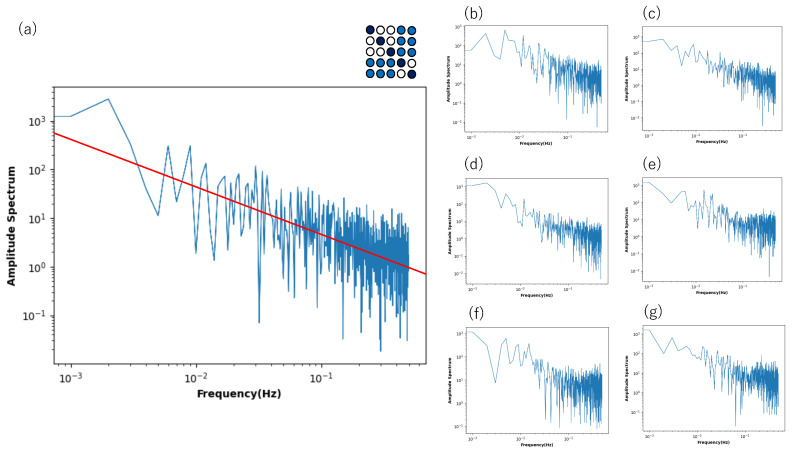
Power spectrum for a time series of the state of a neuron plotted against frequency in a log–log plot. The neuron was chosen from a Boltzmann machine network consisting of 5 × 5 = 25 neurons under the quantum condition, where the connection strength of the non-diagonal sub-relation area was differently given between 0.25 and 0.65 (**a**), 0.2 and 0.7 (**b**), 0.35 and 0.65 (**c**), 0.45 and 0.65 (**d**), 0.1 and 0.2 (**e**), 0.08 and 0.18 (**f**), and 0.09 and 0.19 (**g**). Other connection strength values were defined as in Figure 8. The red line in (**a**) represents a slope of −1.0.

**Figure 12 entropy-27-00145-f012:**
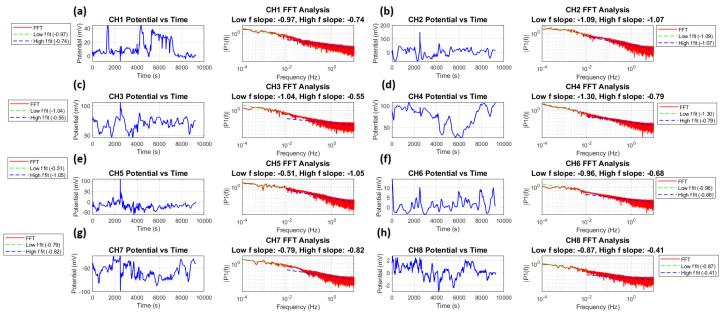
Time series recordings and spectral analysis of Glu-Phe proteinoid potential activity across eight channels (CH1-CH8). (**a**) CH1 recordings show potential fluctuations (±40 mV) with dual-regime FFT analysis revealing slopes of $\beta_{low}=-0.97$ and $\beta_{high}=-0.74$. (**b**) CH2 displays larger amplitude oscillations (±100 mV) with consistent FFT slopes of $\beta_{low}=-1.09$ and $\beta_{high}=-1.07$, indicating robust pink noise characteristics across frequencies. (**c**) CH3 exhibits elevated baseline potentials (60–100 mV) with distinct scaling regimes of $\beta_{low}=-1.04$ and $\beta_{high}=-0.55$. (**d**) CH4 shows sustained high-amplitude activity (40–100 mV) with FFT slopes of $\beta_{low}=-1.30$ and $\beta_{high}=-0.79$. (**e**) CH5 demonstrates balanced oscillations (±50 mV) with contrasting dynamics of $\beta_{low}=-0.51$ and $\beta_{high}=-1.05$. (**f**) CH6 displays lower amplitude fluctuations (±10 mV) with FFT slopes of $\beta_{low}=-0.96$ and $\beta_{high}=-0.68$. (**g**) CH7 shows negative-biased potentials (−40 to −100 mV) with similar scaling in both regimes $\beta_{low}=-0.79$ and $\beta_{high}=-0.82$. (**h**) CH8 exhibits minimal fluctuations (±2 mV) with shallow slopes of $\beta_{low}=-0.87$ and $\beta_{high}=-0.41$. Left panels show spontaneous potential fluctuations over 10,000 s. Right panels show Fast Fourier Transform analyses in log–log scale, revealing distinct power law behaviour ($1/f^{\beta }$) in low- (f≤0.01 Hz) and high- (f>0.01 Hz) frequency regimes. The analysis reveals complex multi-scale dynamics, with CH2 showing the most consistent pink noise characteristics across both frequency regimes (β≈1), while other channels display varying degrees of scale-dependent behaviour.

**Table 1 entropy-27-00145-t001:** Scaling exponent analysis of bioelectric signals from proteinoid microspheres across two frequency regimes. Power spectrum exponents ($\beta_{low}$, $\beta_{high}$) indicate temporal correlations in different frequency ranges. Theoretical scaling exponent ($\alpha_{theo}$) is calculated from relation α=(1+|β|)/2, while $\alpha_{fit}$ is obtained from direct fitting of root mean square fluctuation.

Channel	$\beta_{\mathrm{low}}$	$\beta_{\mathrm{high}}$	$\alpha_{\mathrm{theo},\mathrm{low}}$	$\alpha_{\mathrm{theo},\mathrm{high}}$	$\alpha_{\mathrm{fit},\mathrm{low}}$	$\alpha_{\mathrm{fit},\mathrm{high}}$
1	−0.97	−0.74	0.985	0.87	0.51	0.81
2	−1.09	−1.07	1.045	1.035	0.44	0.90
3	−1.04	−0.55	1.02	0.775	0.47	0.80
4	−1.30	−0.79	1.15	0.895	0.67	0.87
5	−0.51	−1.05	0.755	1.025	0.32	0.86
6	−0.96	−0.68	0.98	0.84	0.46	0.75
7	−0.79	−0.82	0.895	0.91	0.40	0.87
8	−0.87	−0.41	0.935	0.705	0.42	0.64

## Data Availability

The data presented in this study are available on request from the corresponding author.

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
