# Peer review of "Recurrence Resonance and 1/f Noise in Neurons Under Quantum Conditions and Their Manifestations in Proteinoid Microspheres"

_entropy, 2025, doi:10.3390/e27020145_

Round 1

Reviewer 1 Report

Comments and Suggestions for Authors

Equations (9)–(12):  As mentioned in reference [8] and other references regarding Boltzmann machine, the state vector of Boltzmann machine should consist of elements s_i​ (the output of the i-th neuron in a 1D network) or s_{ij}​ (the output of the i-th neuron in a 2D network), rather than elements u_i​ (the total input to the i-th neuron in a 1D network) or u_{ij}​ (the total input to the i-th neuron in a 2D network). Given the use of total input to represent the state of the network, the authors may need to revisit all their theoretical and numerical computations or provide an explanation for this choice.

Fourier series C(f) defined in Equation (13): 

(a) If I understand correctly, the network's time duration is t=0, 1, …, T,  where T represents the end time. Please clarify why the frequency range of C(f) is f=0,1, … T-1.  

(b) please explain why Fourier transformation is computed from t=tau to t=tau+500.

(c) please elaborate on why 1/T is used in the definition of C(f).

(d) Please justify the use of e^{-i(2pi*t*f)/T} in the expression for C(f). Based on the standard definition of the discrete-time Fourier transform, it should typically be e^{-i(2pi*t*f)} assuming t=0,1, …, T, … infinity.

Figures (4)–(8):

(a) The vertical axis labels two measures, entropy and mutual information, but it is difficult to distinguish which curves correspond to entropy and which to mutual information. I suggest using solid and dashed lines to differentiate between the two measures for improved clarity.

(b) Equations (7) and (8) define the entropy of a single neuron and the mutual information between a pair of neurons. Does the vertical axis represent the average entropy across all neurons and the average mutual information across all pairs of neurons?

(c) The labels of horizontal and vertical axis in some figures are hard to read.

Recurrent networks: as mentioned in reference [7], recurrence resonance demonstrates that noise can enhance information flux in recurrent neural networks. In this manuscript, four types of networks (normal, diagonal, blocky, quantum) are examined. Could you please clarify whether these networks are recurrent?

Author Response

our reply is contained in the attachment, "A letter to Reviewer 1"

Reviewer 2 Report

Comments and Suggestions for Authors

The paper did a focused study on the relation between recurrent resonance and 1/f noise, both of which are related to the notion ‘edge of chaos’ in complex systems. It is done in the context of the artificial neural network - Boltzmann machine to be specific. The authors managed find out the cases when single neuron does or does not exhibit recurrent resonance, and gave some intuitive explanations to it. The paper is clearly written with a good discussion of relevant literatures, as well as relatively thorough simulation work.

I have some questions and suggestions to the work.

  1. The authors managed to give a good overview of various signatures of critical phenomena, ranging from SOC, recurrence resonance and quantum automata 1/f noise, and presented some discussion about the open questions. However, there seems to be a lack of ‘so what’ discussion on the study, as to what is the impact of this study brings. Given the detailed discussion in the introduction and conclusion, the authors are close to distilling this point, but still not clear enough.
  2. There are places when the authors mention that there the Boltzmann machine has both chaos and order appearing, but no direct evidence is given. Is it inferred from the peak in mutual information? And if so how are these two connected?
  3. For the fluctuation analysis as alternative way for 1/f noise, from table 1 it seems that the alpha value is calculated straight away from (1+beta)/2, rather than fitting the data using equations 25, is that so? It would be better to estimate it directly from data since it is doable, and relatedly, the ‘detrended fluctuation analysis’ are more commonly used and can in cases yield better results. It is worth seeing how that would affect the analysis.
  4. Related to the above point, the results on proteinoid in figure 12 seems to be not fitting the power spectrum well enough. In particular, the lower frequency regime seem to be flatter than the higher frequency regime. And on a visual inspection, it seems the higher frequencies are closer to exponent -1.0. I would suggest the authors to do the fitting for the two regimes separately.
  5. While the authors draw some connection between the artificial Boltzmann machine and the proteinoid, the connections are rather ‘conceptual’, and at the numerical level do not tell a same story, as the latter can have exponents quite different from -1.0. I guess with more future work closer relationship can be established, but having the proteinoid result in the same paper looks very incoherent at this stage. Therefore it might be advisable to remove the latter from the work. But this is a mild suggestion and I am ok if they authors have a strong reason to keep it.
  6. A minor point: in equation 4, how is the value of eta_ij determined? I might have missed it if it is mentioned somewhere in the manuscript.

Author Response

Our reply is contained in the attachment, " a letter to Reviewer 2".

Round 2

Reviewer 2 Report

Comments and Suggestions for Authors

The authors have addressed my comments mostly, and i am ok to accept it.